# An Outbreak of COVID-19 among mRNA-Vaccinated Nursing Home Residents

**DOI:** 10.3390/vaccines9080859

**Published:** 2021-08-04

**Authors:** Fabrizio Faggiano, Maicol Andrea Rossi, Tiziana Cena, Fulvia Milano, Antonella Barale, Quenya Ristagno, Virginia Silano

**Affiliations:** 1Department of Translational Medicine, University of Piemonte Orientale, 28100 Novara, Italy; fabrizio.faggiano@uniupo.it; 2Epidemiology Unit, ASL Vercelli, 13100 Vercelli, Italy; tiziana.cena@aslvc.piemonte.it (T.C.); antonella.barale@aslvc.piemonte.it (A.B.); 3Lab Unit, ASL Vercelli, 13100 Vercelli, Italy; fulvia.milano@aslvc.piemonte.it; 4Public Health Unit, ASL Vercelli, 13100 Vercelli, Italy; quenya.ristagno@aslvc.piemonte.it (Q.R.); virginia.silano@aslvc.piemonte.it (V.S.)

**Keywords:** COVID-19, SARS-CoV-2, nursing home, long-term care, COVID-19 vaccines

## Abstract

An outbreak was described among the guests of a Long-Term Care Facility in the North of Italy. Among 23 guests, 20 of whom were fully vaccinated with BNT162b2 vaccine, the outbreak led to a final count of 11 positive guests, 9 of whom were vaccinated, and 4 positive healthcare workers, of whom only 1 was vaccinated. Eight of the positive guests (six vaccinated and two unvaccinated) had symptoms that in five cases (three vaccinated and two unvaccinated) led to death. The risk of infection and the risk of death appeared not to be correlated with the health status neither with the serological titer, but only with age.

## 1. Introduction

Long-term care facilities (LTCFs) have been one of the contexts most affected by the COVID-19 pandemic worldwide, due to the particular frailty of their residents, their particular organization (where the socialization aspect plays an important role in the life of the residents), and often critical issues related to care. For this reason, in the first phase of the Italian National Strategic Plan for SARS-CoV-2/COVID-19 Vaccination, the target population prioritized healthcare workers alongside guests and staff of LTCFs [1,2].

The Italian COVID-19 vaccination campaign started on 27 December 2020, a few days after the European Medicines Agency (EMA) approval of the BNT162b2 vaccine to prevent COVID-19 in the EU.

The multinational pivotal efficacy trial of the BNT162b2 vaccine had previously shown that a two-dose regimen at a 21-day interval was sufficient to confer a 95% protection against COVID-19 in people 16 years of age or older. No differences in the level of protection were found based on age, sex, race, ethnicity, obesity, and presence of comorbidities [3].

Phase 3 studies are usually carried out in ideal conditions and randomized controlled trials (RCTs) represent best-case scenarios of vaccine efficacy. However, vaccine efficacy as estimated by RCTs does not always predict vaccine effectiveness and this is particularly true in the case of SARS-CoV-2, where the fundamental understanding of the pathogen is evolving [4].

In Italy a progressive reduction in COVID-19 cases, isolations, hospitalizations and deaths was observed in LTCFs during February-April 2021. This was presumably related to the start of the vaccination campaign [2].

Despite these encouraging data, several outbreaks of COVID 19 among vaccinated subjects have been described, both in Italy and worldwide, and it is important to describe and analyze these cases to better understand whether predisposing conditions for vaccine failure may be present [5,6].

The purpose of this paper is to report an unexpected outbreak of COVID-19 infections among vaccinated guests and staff of a nursing home in the province of Vercelli, Italy.

## 2. Results

On 19 March 2021, an unvaccinated nurse working in the aforementioned facility tested positive for SARS-CoV-2. The molecular test on nasopharyngeal swab was carried out on 17 March 2021, four days after the nurse’s last shift. At that time, the LTCF housed 23 guests and 26 healthcare workers, 20 and 15 of whom, respectively, had been vaccinated. All vaccinated individuals had received two doses of BNT162b2 mRNA COVID-19 vaccine (Comirnaty, BioNTech Manufacturing GmbH, Mainz, Germany) between January and early February 2021.

Once the first case was identified, surveillance protocols were immediately activated, followed by an accurate epidemiological investigation confirming full compliance with all guidelines for the use of personal protective equipment (PPE) and implementation of social distancing.

On the same day (19 March), rapid antigenic nasopharyngeal swabs were performed on all staff and guests of the nursing home, revealing five positive cases among the guests and one further positive case among the staff. In all these cases, molecular swab testing confirmed SARS-CoV-2 positivity and the whole genome sequencing identified the coronavirus as belonging to the lineage B.1.1.7, also known as the “English variant”. Four of the positive guests were vaccinated, while the healthcare worker was not. On the same day, blood samples taken from all positive cases were subjected to chemiluminescent immunoassay (CLIA) for anti-SARS-CoV-2 IgG detection (anti-spike protein antibodies).

Despite the full deployment of the isolation protocols, on 26 March, four other vaccinated guests, one of the three unvaccinated guests, and one vaccinated worker tested positive. On 2 April, another vaccinated guest tested positive.

As shown in Table 1, the outbreak led to a final count of 11 positive guests, 9 of whom were vaccinated, and 4 positive healthcare workers, of whom only 1 was vaccinated. Eight of the positive guests had symptoms that in five cases (three vaccinated and two unvaccinated) led to death. The symptoms of the vaccinated guests are described in Table 2. We can notice that fever, cough, dyspnea, inappetence, and asthenia were the most frequent symptoms. The three dead patients all had severe symptoms with dyspnea, inappetence and asthenia.

In order to identify any co-existing conditions that might have contributed to vaccine failure, we next sought to determine the clinical status of all guests through a questionnaire-based survey using three different scoring systems: Barthel, Short Portable Mental Status Questionnaire (SPMSQ) and Cumulative Illness Rating Scale (CIRS).

The Barthel scale is used to measure performance in Activities of Daily Living (ADL) and in the LTCFs is normally used to assess rehabilitation progress and the residual degree of autonomy [7].

The SPMSQ is a widely used 10-item cognitive screening instrument, whose items test orientation to time and place, memory, current event information (date, day of the week, name of this place, telephone number, date of birth, age, name of current prime minister and previous prime minister, mother’s maiden name), and calculation (subtract 3 s starting with number 20). The total number of errors is computed, and it ranges from 0 to 10 [8].

Lastly, CIRS is a standardized instrument used in the geriatric field to measure the health of the elderly as objectively as possible. It requires the physician to assess and measure the clinical and functional severity of 14 disease categories. A severity value must be defined for each of these diseases, based on the clinical history, the objective examination and the symptoms declared by the patient. The CIRS score, when all sources of medical information are carefully selected and considered, has proven to be an accurate and valid measure of the health status and physical illnesses of the elderly and has wide applicability in the field of geriatric research. [9,10,11]. We used a modified version of CIRS developed and validated in a geriatric residential population that provides a disease severity index (from one to five), obtained from the mean of the individual item scores, and a comorbidity index (from 0 to 14), obtained from the number of items scoring three or more [12,13,14,15].

All questionnaires of scoring systems were filled-in by the medical team of the LCTF. Results of questionnaires are presented in Table 3.

The LTCF housed patients with a median age of 89 years. The median Barthel score was 50.50 points demonstrating severe dependency of the residents in ADL. The SPMSQ median value of two points reveals a normal mental functioning of patients while CIRS scores describe a moderate severity of disease with comorbidities referred on average to 7.5 disease categories.

A non-parametric Wilcoxon signed-rank test was performed to compare median values of negative and positive cases in order to identify the determinants of vaccine failure and lethality among the 20 vaccinated guests. Continuous variables are presented as median and range. Among all parameters analyzed, only age showed a statistically significant difference between infected and non-infected cases (*p* = 0.033). No significant differences were found in Barthel, SPMSQ and CIRS scoring. Results are reported in Table 4.

We also compared the characteristics of the positive vaccinated cases stratified by vital status. Considering the small sample size, we were not able to perform statistical analysis, but we can observe that deceased positive cases seems to be older than survivors. No other differences can be found (Table 5).

Even the guests’ serological status, shown in Table 6, did not allow us to predict the risk of infection after vaccination. Of note, the median IgG levels among the positive cases were paradoxically higher (559 BAU/mL) than those recorded in the negative ones (257 BAU/mL). No differences were found between deceased and survived positive subjects.

## 3. Discussion

The effectiveness of vaccination in elderly and often immunocompromised individuals is a matter of discussion in the scientific community, and several studies are underway to estimate it in LTCFs. Even if a recent study showed that a single-dose SARS-CoV-2 vaccination in older adults living in LTCFs provides substantial protection against infection from 4–7 weeks after vaccination and might reduce SARS-CoV-2 transmission, this observation has to be treated with caution, especially in case of new virus variants [16].

Overall, the results of our investigation showing a SARS-CoV-2 infection rate of 45% percent in a BNT162b2-vaccinated population is a totally unexpected finding, that deserves further scrutiny, especially considering the high percentage of symptomatic cases (67%) and deaths (33%) reported herein.

Considering also a recent report by White et al. [17], the effectiveness of vaccination of nursing home guests should be much higher than that observed in our study.

The fact that all observed cases were infected with the SARS-CoV-2 B.1.1.7 lineage, a variant not considered in the study by White et al. [17], which has a 43–90% higher replication rate than the wild-type strain (the strain of virus that contains no major mutations) and an increased risk of death of 61% [18,19], does not fully account for such an excess of risk of infection and mortality.

Another study estimated effectiveness of the vaccine against any documented infection with the B.1.1.7 variant was 89.5% (95% confidence interval (CI), 85.9 to 92.3) at 14 or more days after the second dose [20]. Although this study is not specifically aimed at an elderly population in a residential setting, the observed difference in effectiveness is still high.

Another reason for concern is the lack of correlation between the risk of infection and the anti-SARS-CoV-2 IgG antibody titer. Indeed, given that the serological screening had been performed no later than a week after the detection of the first positive case, we can assume that the IgG titer levels were attributable to the vaccine instead of the infection. Under this assumption, we would have expected an indirect correlation between the risk of infection and the IgG antibody titer [21].

We also show that the clinical conditions do not correlate with the risk of infection or severe disease, and that the only variable that can explain the excess of risk is old age.

Although our findings are limited by the small sample size of our study population and should, therefore, be regarded as anecdotal evidence, they represent the first warning that the BNT162b2 vaccine effectiveness may be lower than expected in people older than 80 years infected with the B.1.1.7 variant. Thus, we recommend that older people, especially when hosted in LTCFs, even if vaccinated, should be prevented from being exposed to unvaccinated healthcare workers. In addition, as the risk of infection is not eliminated, non-pharmaceutical interventions to reduce viral transmission remain crucial in the management of LTCFs [16].

We also want to highlight the need for ongoing vaccination programs and surveillance testing in LTCFs to mitigate future outbreaks.

## Figures and Tables

**Table 1 vaccines-09-00859-t001:** Summary of outcomes of vaccinated and unvaccinated residents and healthcare workers who tested positive for SARS-CoV-2.

		Total		Positive Cases		Symptomatic		Dead	
**Guests**	**Vaccinated**	20	87.0%	9	45.0%	6	66.7%	3	33.3%
	**Unvaccinated**	3	13.0%	2	66.7%	2	100%	2	100%
	**Total**	23		11		8		5	
**Healthcare workers**	**Vaccinated**	15	57.7%	1	6.7%	0	-	0	-
	**Unvaccinated**	11	42.3%	3	27.3%	1	33.3%	0	-
	**Total**	26		4		1		0	

**Table 2 vaccines-09-00859-t002:** Clinical and serological conditions of the 9 positive vaccinated guests.

Age	IgG (BAU/mL)	Positive Swab Date	Symptoms	Symptom Onset Date	Death
99	260	20 March 2021	Dyspnea; Inappetence; Asthenia; Muscle pain;	23 March 2021	YES
96	2019	20 March 2021	Fever; Dyspnea; Inappetence; Asthenia; Muscle pain;	22 March 2021	YES
95	>2080	20 March 2021	Dyspnea;	1 April 21	NO
94	1000	20 March 2021	-------	-------	NO
90	559	26 March 2021	Fever; Cough; Dyspnea; Asthenia;	29 March 2021	YES
99	<15	26 March 2021	-------	-------	NO
81	<15	26 March 2021	Fever; Cough; Sore throat;	23 March 2021	NO
92	482	26 March 2021	Fever; Rhinitis; Asthenia;	27 March 2021	NO
86	652	3 April 2021	-------	-------	NO

**Table 3 vaccines-09-00859-t003:** Characteristics of the 20 residents vaccinated with the BNT162b2 vaccine (Comirnaty).

Variables	Median	(Min–Max)
Age	89.00	71.0–99.0
Barthel Score	50.50	7.0–60.0
SPMSQ	2.00	0–2.00
CIRS—14 item severity Index	2.37	1.50–3.36
CIRS—14 item comorbidity Index	7.50	3.00–11.00

**Table 4 vaccines-09-00859-t004:** Characteristics of the 20 residents vaccinated with the BNT162b2 vaccine (Comirnaty) stratified by COVID-19 test results.

Variables	PCR Tests				
	Negative Cases (*n* = 11)		Positive Cases (*n* = 9)		*p*-Value
	Median	(Min–Max)	Median	(Min–Max)	
Age	88.0	71.0–98.0	94.0	81.0–99.0	0.033
Barthel Score	45.0	7.0–60.0	55.0	25.0–60.0	0.074
SPMSQ	2.00	0–2.00	2.00	0–2.00	0.814
CIRS—14 item severity Index	2.36	1.50–3.36	2.43	1.93–2.93	0.940
CIRS—14 item comorbidity Index	8.00	3.00–11.00	7.00	3.00–9.00	0.757

**Table 5 vaccines-09-00859-t005:** Characteristics of the BNT162b2-vaccinated guests testing positive for SARS-CoV-2 stratified by vital status.

Variables	Vital Status			
	Alive (*n* = 6)		Dead (*n* = 3)	
	Median	(Min–Max)	Median	(Min–Max)
Age	93.0	81.0–99.0	96.0	90.0–99.0
Barthel Score	55.0	25.0–60.0	55.0	46.0–58.0
SPMSQ	2.00	0–2.00	2.00	1.00–2.00
CIRS—14 item Severity Index	2.29	1.93–2.64	2.43	2.00–2.93
CIRS—14 item Comorbidity Index	7.00	3.00–9.00	8.00	6.00–9.00

**Table 6 vaccines-09-00859-t006:** Anti-SARS-CoV-2 IgG testing of the 20 guests vaccinated with the BNT162b2 mRNA COVID-19 vaccine (Comirnaty). The serological test is considered positive if IgG ≥ 33.8 BAU/mL.

Test	*n*	IgG (BAU/mL)	(Min–Max)
Negative	11	257.0	118.0–996.0
Positive	9	559.0	15.0–2080.0
-Dead	3	559.0	260.0–2019.0
-Alive	6	567.0	15.0–2080.0

## Data Availability

Individual data are not accessible because they’re property of ASL Vercelli.

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
