# Peer review of "An Outbreak of COVID-19 among mRNA-Vaccinated Nursing Home Residents"

_vaccines, 2021, doi:10.3390/vaccines9080859_

Round 1
Reviewer 1 Report
Thank you for allowing me to review the case report entitled, “An outbreak of COVID-19 among mRNA-vaccinated nursing home residents” by Faggiano et al. I believe it presents important information about responsiveness to COVID vaccination in a particularly vulnerable population.
Minor Changes:
- Line 12: “20 of whom fully vaccinated” should read “20 of whom were fully vaccinated”
- Line 26: “prioritize” should read prioritized”
- Line 39:”is matter of discussion” should read “is a matter of discussion”
- Lines 39-47: This seems like it might fit better into a discussion than as a part of the introduction.
- Line 65: “once identified the first case” should read “once the first case was identified”
- Line 70: when you say on the same day, do you mean March 19? I would clarify
- Line 70: “performed to all staff” should read “performed on all staff”
- Line 70: “confirmed the SARS-CoV-2” should read “confirmed SARS-coV-2”
- Line 74: you say that 4 of the positive guests were vaccinated. How long had it been between vaccination and infection?
- Line 121: Would consider an alternative work for “resumed”. Maybe “presented”
- Lines 173-177: Would consider reformatting to be clearer.
- Line 179: consider say “vaccination of nursing home guests”. Rather than “vaccination of nursing homes.”
- Line 188: would consider removing the work “too”
Author Response
Dear reviewer,
thank you for your comments and suggestions.
Response to Reviewer 1 Comments
Point 1: Line 12: “20 of whom fully vaccinated” should read “20 of whom were fully vaccinated”
Thank you, I have made the correction
Point 2: Line 26: “prioritize” should read prioritized”
Thank you, I have made the correction
Point 3: Line 39:”is matter of discussion” should read “is a matter of discussion”
Thank you, I have made the correction
Point 4: Lines 39-47: This seems like it might fit better into a discussion than as a part of the introduction.
I moved the text in the discussion section
Point 5: Line 65: “once identified the first case” should read “once the first case was identified”
Thank you, I have made the correction
Point 6: Line 70: when you say on the same day, do you mean March 19? I would clarify
Yes, it’s March 19. I specified it in the text
Point 7: Line 70: “performed to all staff” should read “performed on all staff”
Thank you, I have made the correction
Point 8: Line 70: “confirmed the SARS-CoV-2” should read “confirmed SARS-coV-2”
Thank you, I have made the correction
Point 9: Line 74: you say that 4 of the positive guests were vaccinated. How long had it been between vaccination and infection?
It is specified in lines 64-66. All the vaccinated guests received the vaccination in January and early February
Point 10: Line 121: Would consider an alternative work for “resumed”. Maybe “presented”
Thank you, I have made the correction
Point 11: Lines 173-177: Would consider reformatting to be clearer.
Moving before these lines the text cited on point 4 I think this section is clearer.
Point 12: Line 179: consider say “vaccination of nursing home guests”. Rather than “vaccination of nursing homes.”
Thank you, I have made the correction
Point 13: Line 188: would consider removing the work “too”
Thank you, I have made the correction
Best regards
Maicol Rossi
Reviewer 2 Report
I have few minor comments to revise before accepting this case report for publication.
Line 35: Please explain the abbreviation RCTs
Line 73: What was the procedure used for identification of English variant? Was it from whole genome sequencing? Or RT-PCR specific for English variant?
Line 87: Is it 3 positive healthcare workers or 4? Table 1 total indicates 3.
Line 181: Reproduction rate? I suggest to use replication rate
Line 182: What is the wild-type strain?
Reference section needs formatting.
Author Response
Dear reviewer,
thank you for your comments and suggestions.
Response to Reviewer 2 Comments
Point 1: Line 35: Please explain the abbreviation RCTs
I inserted the explanation of the abbreviation
Point 2: Line 73: What was the procedure used for identification of English variant? Was it from whole genome sequencing? Or RT-PCR specific for English variant?
It was from whole genome sequencing. I modified the text
Point 3: Line 87: Is it 3 positive healthcare workers or 4? Table 1 total indicates 3.
I’m sorry for the mistake in table 1. The healthcare workers were 4.
Point 4: Line 181: Reproduction rate? I suggest to use replication rate
Thank you, I have made the correction
Point 5: Line 182: What is the wild-type strain?
I clarified in the text
Point 6: Reference section needs formatting.
I checked and formatted were needed
Best regards
Maicol Rossi